# Query-Efficient Hard-label Black-box Attack: An Optimization-based Approach

**Minhao Cheng, Huan Zhang & Cho-Jui Hsieh**
Department of Computer Science
University of California, Los Angeles
{mhcheng,huanzhang,chohsieh}@cs.ucla.edu

**Thong Le**
Department of Computer Science
University of California, Davis
thmle@ucdavis.edu

**Pin-Yu Chen**
IBM Research AI
pin-yu.chen@ibm.com

**Jinfeng Yi**
JD AI Research
yijinfeng@jd.com

## Abstract

We study the problem of attacking machine learning models in the *hard-label black-box* setting, where no model information is revealed except that the attacker can make queries to probe the corresponding hard-label decisions. This is a very challenging problem since the direct extension of state-of-the-art white-box attacks (e.g., C&W or PGD) to the hard-label black-box setting will require minimizing a non-continuous step function, which is combinatorial and cannot be solved by a gradient-based optimizer. The only two current approaches are based on random walk on the boundary (Brendel et al., 2017) and random trials to evaluate the loss function (Ilyas et al., 2018), which require lots of queries and lacks convergence guarantees. We propose a novel way to formulate the hard-label black-box attack as a real-valued optimization problem which is usually continuous and can be solved by any zeroth order optimization algorithm, such as randomized gradient-free method (Nesterov & Spokoiny, 2017). We demonstrate that our proposed method outperforms the previous stochastic approaches to attacking convolutional neural networks on MNIST, CIFAR, and ImageNet datasets. More interestingly, the proposed algorithm can also be used to attack other discrete and non-continuous machine learning models, such as Gradient Boosting Decision Trees.

## 1 Introduction

It has been observed recently that machine learning algorithms, especially deep neural networks, are vulnerable to adversarial examples (Goodfellow et al., 2014; Szegedy et al., 2013; Moosavi-Dezfooli et al.; Moosavi Dezfooli et al., 2016; Chen et al., 2018a; Cheng et al., 2018). For example, in image classification problems, attack algorithms (Carlini & Wagner, 2017; Goodfellow et al., 2014; Chen et al., 2017) can find adversarial examples for almost every image with very small human-imperceptible perturbation. The problem of finding an adversarial example can be posed as solving an optimization problem—within a small neighbourhood around the original example, find a point to optimize the cost function measuring the "successfulness" of an attack. Solving this objective function with gradient-based optimizer leads to state-of-the-art attacks (Carlini & Wagner, 2017; Goodfellow et al., 2014; Chen et al., 2017; Szegedy et al., 2013; Madry et al., 2018).

Most current attacks (Goodfellow et al., 2014; Carlini & Wagner, 2017; Szegedy et al., 2013; Chen et al., 2018b) consider the "white-box" setting, where the machine learning model is fully exposed to the attacker. In this setting, the gradient of an attacking objective function can be computed by back-propagation, so attacks can be done very easily. This white-box setting is clearly unrealistic when the model parameters are unknown to an attacker. Instead, several recent works consider the "score-based black-box" setting, where the machine learning model is unknown to the attacker, but it is possible to make queries to obtain the corresponding probability outputs of the model (Chen et al., 2017; Ilyas et al., 2018). However, in many cases real-world models will not provide probability outputs to users. Instead, only the final decision (e.g., top-1 predicted class) can be observed. It is therefore interesting to show whether machine learning model is vulnerable in this setting.

Furthermore, existing gradient-based attacks cannot be applied to some non-continuous machine learning models which involve discrete decisions. For example, the robustness of decision-tree based models (random forest and gradient boosting decision trees (GBDT)) cannot be evaluated using gradient-based approaches, since the gradient of these functions may not exist.

In this paper, we develop an optimization-based framework for attacking machine learning models in a more realistic and general "hard-label black-box" setting. We assume that the model is not revealed and the attacker can only make queries to acquire the corresponding **hard-label decision** instead of the probability outputs (also known as soft labels).

We make hard-label black-box attacks query-efficient by reformulating the attack as a novel real-valued optimization problem, which is usually continuous and much easier to solve. Although the objective function of this reformulation cannot be written in an analytical form, we show how to use (hard-label) model queries to evaluate its function value and thus any zeroth order optimization algorithm can be applied to solve it. In the experiments, we show our algorithm can be successfully used to attack hard-label black-box CNN models on MNIST, CIFAR, and ImageNet with far less number of queries compared to the state-of-art algorithm both in $L_2$ and $L_\infty$ metric.

Moreover, since our algorithm does not depend on the gradient of the classifier, we can apply it to attack other non-differentiable classifiers besides neural networks. We show an interesting application in attacking Gradient Boosting Decision Tree, which cannot be attacked by all the existing gradient-based methods even in the white-box setting. Our method can successfully find adversarial examples with imperceptible perturbations for a GBDT within 30,000 queries.

## 2 BACKGROUND AND RELATED WORK

We will first introduce our problem setting and give a brief literature review to highlight the difficulty of attacking hard-label black-box models.

**Problem Setting**   For simplicity, we consider attacking a $K$-way multi-class classification model in this paper. Given the classification model $f : \mathbb{R}^d \to \{1, \ldots, K\}$ and an original example $\boldsymbol{x}_0$, the goal is to generate an adversarial example $\boldsymbol{x}$ such that

$$\boldsymbol{x} \text{ is close to } \boldsymbol{x}_0 \quad \text{and} \quad f(\boldsymbol{x}) \neq f(\boldsymbol{x}_0) \quad (\text{i.e., } \boldsymbol{x} \text{ is misclassified by model } f.) \tag{1}$$

**White-box attacks**   Most attack algorithms in the literature consider the white-box setting, where the classifier $f$ is exposed to the attacker. For neural networks, under this assumption, back-propagation can be conducted on the target model because both network structure and weights are known by the attacker. For classification models in neural networks, it is usually assumed that $f(\boldsymbol{x}) = \operatorname{argmax}_i(Z(\boldsymbol{x})_i)$, where $Z(\boldsymbol{x}) \in \mathbb{R}^K$ is the final (logit) layer output, and $Z(\boldsymbol{x})_i$ is the prediction score for the $i$-th class. The objectives in (1) can then be naturally formulated as the following optimization problem:

$$\operatorname*{argmin}_{\boldsymbol{x}} \left\{ \operatorname{Dis}(\boldsymbol{x}, \boldsymbol{x}_0) + c\mathcal{L}(Z(\boldsymbol{x})) \right\} := h(\boldsymbol{x}), \tag{2}$$

where $\operatorname{Dis}(\cdot, \cdot)$ is some distance measurement (e.g., $L_2, L_1$ or $L_\infty$ norm in Euclidean space), $\mathcal{L}(\cdot)$ is the loss function corresponding to the goal of the attack, and $c$ is a balancing parameter. For *untargeted attack*, where the goal is to make the target classifier misclassify, the loss function can be defined as

$$\mathcal{L}(Z(\boldsymbol{x})) = \max\{[Z(\boldsymbol{x})]_{y_0} - \max_{i \neq y_0}[Z(\boldsymbol{x})]_i, -\kappa\}, \tag{3}$$

where $y_0$ is the original label predicted by the classifier, $\kappa$ is the margin (usually set to be 1 or 0) of the hinge loss. For *targeted attack*, where the goal is to turn it into a specific target class $t$, the loss function can also be defined accordingly.

Therefore, attacking a machine learning model can be posed as solving this optimization problem (Carlini & Wagner, 2017; Chen et al., 2018b), which is also known as the C&W attack or the EAD attack depending on the choice of the distance measurement. To solve (2), one can apply any gradient-based optimization algorithm such as SGD or Adam, since the gradient of $\mathcal{L}(Z(\boldsymbol{x}))$ can be computed via back-propagation.

The ability of computing gradient also enables many different attacks in the white-box setting. For example, eq (2) can also be turned into a constrained optimization problem, which can then

be solved by projected gradient descent (PGD) (Madry et al., 2018). FGSM (Goodfellow et al., 2014) is the special case of one step PGD with $L_\infty$ norm distance. Other algorithms such as Deepfool (Moosavi Dezfooli et al., 2016) also solve similar optimization problems to construct adversarial examples.

**Previous work on black-box attack**     In real-world systems, usually the underlying machine learning model will not be revealed and thus white-box attacks cannot be applied. This motivates the study of attacking machine learning models in the *black-box setting*, where attackers do not have any information about the function $f$. And the only valid operation is to make queries to the model and acquire the corresponding output $f(\boldsymbol{x})$. The first approach for black-box attack is using transfer attack (Papernot et al., 2017) – instead of attacking the original model $f$, attackers try to construct a substitute model $\hat{f}$ to mimic $f$ and then attack $\hat{f}$ using white-box attack methods. This approach has been well studied and analyzed in (Liu et al., 2017; Bhagoji et al., 2017). However, recent papers have shown that attacking the substitute model usually leads to much larger distortion and low success rate (Chen et al., 2017). Therefore, instead, (Chen et al., 2017) considers the *score-based* black-box setting, where attackers can use $\boldsymbol{x}$ to query the softmax layer output in addition to the final classification result. In this case, they can reconstruct the loss function (3) and evaluate it as long as the objective function $h(\boldsymbol{x})$ exists for any $\boldsymbol{x}$. Thus a zeroth order optimization approach can be directly applied to minimize $h(\boldsymbol{x})$. (Tu et al., 2018) further improves the query complexity of (Chen et al., 2017) by introducing an autoencoder-based approach to reduce query counts and an adaptive random gradient estimation to balance query counts and distortion.

**Difficulty of hard-label black-box attacks**     Throughout this paper, the hard-label black-box setting refers to cases where real-world ML systems only provide limited prediction results of an input query. Specifically, only the final decision (top-1 predicted label) instead of probability outputs is known to an attacker.

Attacking in this setting is indeed very challenging. In Figure 1a, we show a simple 3-layer neural network's decision boundary. Note that the $\mathcal{L}(Z(\boldsymbol{x}))$ term is continuous as in Figure 1b because the logit layer output is real-valued functions. However, in the hard-label black-box setting, only $f(\cdot)$ is available instead of $Z(\cdot)$. Since $f(\cdot)$ can only be a one-hot vector, if we plug-in $f$ into the loss function, $\mathcal{L}(f(\boldsymbol{x}))$ (as shown in Figure 1c) will be discontinuous and with discrete outputs.

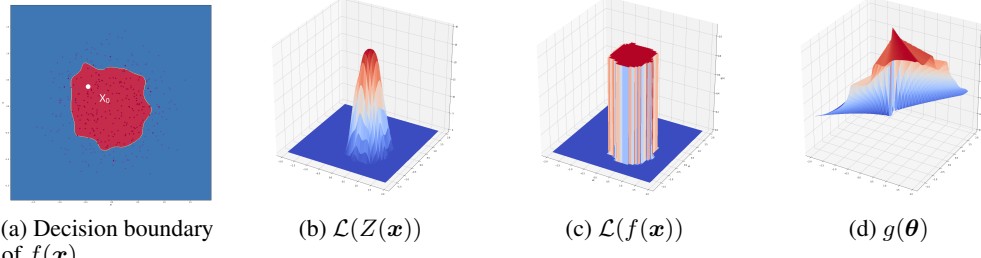

(a) Decision boundary of $f(\boldsymbol{x})$          (b) $\mathcal{L}(Z(\boldsymbol{x}))$          (c) $\mathcal{L}(f(\boldsymbol{x}))$          (d) $g(\boldsymbol{\theta})$

Figure 1: (a) A neural network classifier. (b) illustrates the loss function of C&W attack, which is continuous and hence can be easily optimized. (c) is the C&W loss function in the hard-label setting, which is discrete and discontinuous. (d) our proposed attack objective $g(\boldsymbol{\theta})$ for this problem, which is continuous and easier to optimize. See detailed discussions in Section 3.

Optimizing this function will require combinatorial optimization or search algorithms, which is challenging given the high dimensionality of the problem. The only two current approaches (Brendel et al., 2017; Ilyas et al., 2018) are based on random-walk on the boundary and random trails on the loss function. Although these "Boundary attack" and "Limited attack" can find adversarial examples with comparable distortion with white-box attacks, they need lots of queries to explore the high-dimensional space and lack convergence guarantees. We show that our optimization-based algorithm can significantly reduce the number of queries, and has guaranteed convergence in the number of iterations (queries) when the objective function is lipschitz smooth.

## 3   ALGORITHMS

Now we introduce a novel way to re-formulate hard-label black-box attack as another optimization problem, show how to evaluate the function value using hard-label queries, and then apply a zeroth order optimization algorithm to solve it.

### 3.1 A Boundary-based Re-formulation

For a given example $x_0$, true label $y_0$ and the hard-label black-box function $f : \mathbb{R}^d \to \{1, \ldots, K\}$, we define our objective function $g : \mathbb{R}^d \to \mathbb{R}$ depending on the type of attack:

$$\textbf{Untargeted attack:}\quad g(\boldsymbol{\theta}) = \min_{\lambda > 0} \lambda \quad \texttt{s.t} \quad f(x_0 + \lambda \frac{\boldsymbol{\theta}}{\|\boldsymbol{\theta}\|}) \neq y_0 \qquad (4)$$

$$\textbf{Targeted attack (given target } t\textbf{):}\quad g(\boldsymbol{\theta}) = \min_{\lambda > 0} \lambda \quad \texttt{s.t} \quad f(x_0 + \lambda \frac{\boldsymbol{\theta}}{\|\boldsymbol{\theta}\|}) = t \qquad (5)$$

In this formulation, $\boldsymbol{\theta}$ represents the search direction and $g(\boldsymbol{\theta})$ is the distance from $x_0$ to the nearest adversarial example along the direction $\boldsymbol{\theta}$. The difference between (4) and (5) corresponds to the different definitions of "successfulness" in untargeted and targeted attack, where the former one aims to turn the prediction into any incorrect label and the later one aims to turn the prediction into the target label. For untargeted attack, $g(\boldsymbol{\theta})$ also corresponds to the distance to the decision boundary along the direction $\boldsymbol{\theta}$. In image problems the input domain of $f$ is bounded, so we will impose corresponding upper/lower bounds in the definition of (4) and (5).

Instead of searching for an adversarial example, we search the direction $\boldsymbol{\theta}$ to minimize the distortion $g(\boldsymbol{\theta})$, which leads to the following optimization problem:

$$\min_{\boldsymbol{\theta}} \ g(\boldsymbol{\theta}). \qquad (6)$$

Finally, the adversarial example can be found by $x^* = x_0 + g(\boldsymbol{\theta}^*) \frac{\boldsymbol{\theta}^*}{\|\boldsymbol{\theta}^*\|}$, where $\boldsymbol{\theta}^*$ is the optimal solution of (6).

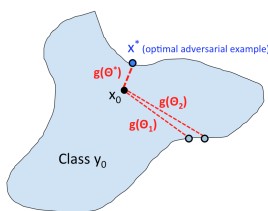

Note that unlike the C&W or PGD objective functions, which are discontinuous step functions in the hard-label setting (see Section 2), $g(\boldsymbol{\theta})$ maps input direction to real-valued output (distance to decision boundary), which is usually continuous – a small change of $\boldsymbol{\theta}$ usually leads to a small change of $g(\boldsymbol{\theta})$, as can be seen from Figure 2.

Figure 2: Illustration

Moreover, we give three examples of $f(x)$ defined in two dimension input space and their corresponding $g(\boldsymbol{\theta})$. In Figure 3a, we have a continuous classification function defined as follows

$$f(x) = \begin{cases} 1, & \text{if } \|x\|_2^2 \geq 0.4 \\ 0, & \text{otherwise.} \end{cases}$$

In this case, as shown in Figure 3c, $g(\boldsymbol{\theta})$ is continuous. Moreover, in Figure 3b and Figure 1a, we show decision boundaries generated by GBDT and neural network classifier, which are not continuous. However, as showed in Figure 3d and Figure 1d, even if the classifier function is not continuous, $g(\boldsymbol{\theta})$ is still continuous. This makes it easy to apply zeroth order method to solve (6).

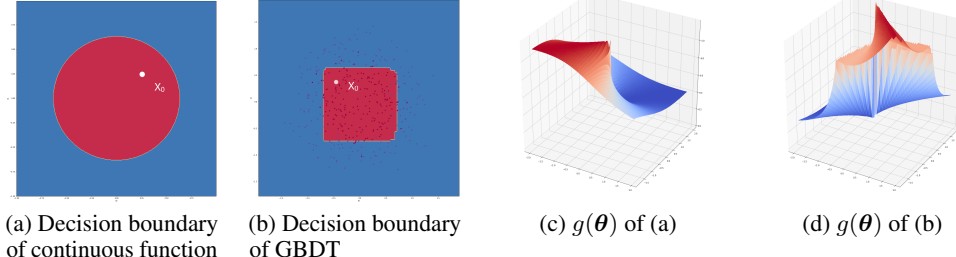

| (a) Decision boundary of continuous function | (b) Decision boundary of GBDT | (c) $g(\boldsymbol{\theta})$ of (a) | (d) $g(\boldsymbol{\theta})$ of (b) |

Figure 3: Examples of decision boundary of classification function $f(x)$ and corresponding $g(\boldsymbol{\theta})$.

**Compute $g(\boldsymbol{\theta})$ up to certain accuracy.** We are not able to evaluate the gradient of $g$, but we can evaluate the function value of $g$ using the hard-label queries to the original function $f$. For simplicity, we focus on untargeted attack here, but the same procedure can be applied to targeted attack as well.

First, we discuss how to compute $g(\boldsymbol{\theta})$ directly without additional information. This is used in the initialization step of our algorithm. For a given normalized $\boldsymbol{\theta}$, we do a coarse-grained search and then a binary search. In coarse-grained search, we query the points $\{x_0 + \alpha\boldsymbol{\theta}, x_0 + 2\alpha\boldsymbol{\theta}, \ldots\}$ one by one until we find $f(x + i\alpha\boldsymbol{\theta}) \neq y_0$. This means the boundary lies between $[x_0 + (i-1)\alpha\boldsymbol{\theta}, x_0 + i\alpha\boldsymbol{\theta}]$. We then enter the second phase and conduct a binary search to find the solution within this region (same with line 11–17 in Algorithm 1). Note that there is an upper bound of the first stage if we

choose $\boldsymbol{\theta}$ by the direction of $\boldsymbol{x} - \boldsymbol{x}_0$ with some $\boldsymbol{x}$ from another class. This procedure is used to find the initial $\boldsymbol{\theta}_0$ and corresponding $g(\boldsymbol{\theta}_0)$ in our optimization algorithm. We omit the detailed algorithm for this part since it is similar to Algorithm 1.

Next, we discuss how to compute $g(\boldsymbol{\theta})$ when we know the solution is very close to a reference point $v$. This is used in all the function evaluations in our optimization algorithm, since the current solution is usually close to the previous solution, and when we estimate the gradient using (7), the queried direction will only be a slight modification of the previous one. In this case, we first increase or decrease $v$ in the local region to find the interval that contains the nearby boundary (e.g, $f(\boldsymbol{x}_0 + v\boldsymbol{\theta}) = y_0$ and $f(\boldsymbol{x}_0 + v'\boldsymbol{\theta}) \neq y_0$), then conduct a binary search to find the final value of $g$. Our procedure for computing the $g$ value is presented in Algorithm 1.

---

**Algorithm 1** Compute $g(\boldsymbol{\theta})$ locally

1: **Input:**  Hard-label model $f$, original image $x_0$, query direction $\boldsymbol{\theta}$, previous solution $v$, increase/decrease ratio $\alpha = 0.01$, stopping tolerance $\epsilon$ (maximum tolerance of computed error)
2: $\boldsymbol{\theta} \leftarrow \boldsymbol{\theta}/\|\boldsymbol{\theta}\|$
3: **if** $f(\boldsymbol{x}_0 + v\boldsymbol{\theta}) = y_0$ **then**
4:     $v_{left} \leftarrow v, v_{right} \leftarrow (1 + \alpha)v$
5:     **while** $f(\boldsymbol{x}_0 + v_{right}\boldsymbol{\theta}) = y_0$ **do**
6:         $v_{right} \leftarrow (1 + \alpha)v_{right}$
7: **else**
8:     $v_{right} \leftarrow v, v_{left} \leftarrow (1 - \alpha)v$
9:     **while** $f(\boldsymbol{x}_0 + v_{left}\boldsymbol{\theta}) \neq y_0$ **do**
10:        $v_{left} \leftarrow (1 - \alpha)v_{left}$
11: ## Binary Search within $[v_{left}, v_{right}]$
12: **while** $v_{right} - v_{left} > \epsilon$ **do**
13:     $v_{mid} \leftarrow (v_{right} + v_{left})/2$
14:     **if** $f(\boldsymbol{x}_0 + v_{mid}\boldsymbol{\theta}) = y_0$ **then**
15:         $v_{left} \leftarrow v_{mid}$
16:     **else**
17:         $v_{right} \leftarrow v_{mid}$
18: **return** $v_{right}$

---

### 3.2 Hard-label Black-box Attacks with $L_\infty$ norm constraint

Although we could let $\|\boldsymbol{\theta}\| = \|\boldsymbol{\theta}\|_\infty$ in (4) and (5) directly, $g(\boldsymbol{\theta})$ will be harder to optimize in practice because of introducing the $\max$ term in $\|\cdot\|_\infty$. Instead, with an $L_\infty$ constraint $\varepsilon$, we design a smooth approximation loss as follows:

**Untargeted attack:**  $g(\boldsymbol{\theta}) = \min_\lambda\{\sum_{i=1}^{d}(\max\{\lambda\frac{|\boldsymbol{\theta}_i|}{\|\boldsymbol{\theta}\|_\infty} - \varepsilon, 0\})^2\}$   s.t   $f(\boldsymbol{x}_0 + \lambda\frac{\boldsymbol{\theta}}{\|\boldsymbol{\theta}\|_\infty}) \neq y_0$

**Targeted attack:**  $g(\boldsymbol{\theta}) = \min_\lambda\{\sum_{i=1}^{d}(\max\{\lambda\frac{|\boldsymbol{\theta}_i|}{\|\boldsymbol{\theta}\|_\infty} - \varepsilon, 0\})^2\}$   s.t   $f(\boldsymbol{x}_0 + \lambda\frac{\boldsymbol{\theta}}{\|\boldsymbol{\theta}\|_\infty}) = t.$

Here $\boldsymbol{\theta}_i$ is the i-th coordinate of $\boldsymbol{\theta}$. Notably, when $\lambda \leq \varepsilon$, we have $g(\boldsymbol{\theta}) = 0$. That's to say, we have obtained a legitimate $\boldsymbol{\theta}$ to make a valid adversarial example $\boldsymbol{x}_0 + \lambda^*\frac{\boldsymbol{\theta}}{\|\boldsymbol{\theta}\|_\infty}$.

### 3.3 Zeroth Order Optimization

To solve the optimization problem (6) for which we can only evaluate function value instead of gradient, zeroth order optimization algorithms can be naturally applied. In fact, after the reformulation, the problem can be potentially solved by any zeroth order optimization algorithm, like zeroth order gradient descent, genetic algorithm (Alzantot et al., 2018) or coordinate descent (see Conn et al. (2009) for a comprehensive survey).

Here we propose to solve (1) using Randomized Gradient-Free (RGF) method proposed in (Nesterov & Spokoiny, 2017; Ghadimi & Lan, 2013). In practice, we found it outperforms zeroth-order coordinate descent. At each iteration, the gradient is estimated by

$$\hat{\boldsymbol{g}} = \frac{g(\boldsymbol{\theta} + \beta\boldsymbol{u}) - g(\boldsymbol{\theta})}{\beta} \cdot \boldsymbol{u} \tag{7}$$

---

**Algorithm 2** RGF for hard-label black-box attack

---

1: **Input:** Hard-label model $f$, original image $x_0$, initial $\boldsymbol{\theta}_0$.
2: **for** $t = 0, 1, 2, \ldots, T$ **do**
3:     Randomly choose $\boldsymbol{u}_t$ from a zero-mean Gaussian distribution
4:     Evaluate $g(\boldsymbol{\theta}_t)$ and $g(\boldsymbol{\theta}_t + \beta \boldsymbol{u})$ using Algorithm 1
5:     Compute    $\hat{\boldsymbol{g}} = \dfrac{g(\boldsymbol{\theta}_t + \beta \boldsymbol{u}) - g(\boldsymbol{\theta}_t)}{\beta} \cdot \boldsymbol{u}$
6:     Update    $\boldsymbol{\theta}_{t+1} = \boldsymbol{\theta}_t - \eta_t \hat{\boldsymbol{g}}$
7: **return** $\boldsymbol{x}_0 + g(\boldsymbol{\theta}_T)\boldsymbol{\theta}_T$

---

where $\boldsymbol{u}$ is a random Gaussian vector, and $\beta > 0$ is a smoothing parameter (we set $\beta = 0.005$ in all our experiments). The solution is then updated by $\boldsymbol{\theta} \leftarrow \boldsymbol{\theta} - \eta \hat{\boldsymbol{g}}$ with a step size $\eta$. The procedure is summarized in Algorithm 2.

Also, if $g(\boldsymbol{\theta})$ is Lipschitz-smooth, we are able to bound the number of iterations needed with $O(\frac{d}{\delta^2})$ for our algorithm to achieve stationary points. See the appendix for details.

### 3.3.1 IMPLEMENTATION DETAILS

There are several implementation details when we apply this algorithm. First, for high-dimensional problems, we found the estimation in (7) is very noisy. Therefore, instead of using one vector, we sample $q$ vectors from Gaussian distribution and average their estimators to get $\hat{\boldsymbol{g}}$. We set $q = 20$ in all the experiments. Second, instead of using a fixed step size (suggested in theory), we use a backtracking line-search approach to find step size at each step. This leads to additional query counts, but makes the algorithm more stable and eliminates the need to hand-tuning the step size. Third, instead of using a random direction $\boldsymbol{\theta}$ as initialization, we sample $t$ vectors from Gaussian distribution and choose the one with smallest $g(\boldsymbol{\theta})$ as our initialization. It helps us to find a good initialization direction and thus get a smaller distortion in the end with limited number of additional queries. We set $t = 100$ in all the experiments.

## 4 EXPERIMENTAL RESULTS

We test the performance of our hard-label black-box attack algorithm on convolutional neural network (CNN) models and compare with Boundary attack (Brendel et al., 2017), Limited attack (Ilyas et al., 2018) and a random trail baseline. Furthermore, we show our method can be applied to attack Gradient Boosting Decision Tree (GBDT) and present some interesting findings.

### 4.1 ATTACKING CNN IMAGE CLASSIFICATION MODELS

We use three standard datasets: MNIST (LeCun et al., 1998), CIFAR-10 (Krizhevsky, 2009) and ImageNet-1000 (Deng et al., 2009). To have a fair comparison with previous work, we adopt the same networks used in both Carlini & Wagner (2017) and Brendel et al. (2017). In detail, both MNIST and CIFAR use the same network structure with four convolution layers, two max-pooling layers and two fully-connected layers. Using the parameters provided by Carlini & Wagner (2017), we could achieve 99.5% test accuracy on MNIST and 82.5% test accuracy on CIFAR-10, which is similar to accuracy reported in Carlini & Wagner (2017). For Imagenet-1000, we use the pretrained network Resnet-50 (He et al., 2016) and Inception-V3 (Szegedy et al., 2016) provided by torchvision[1], which could achieve 76.15% and 77.45 % top-1 test accuracy respectively. For simplicity, all images are normalized into $[0, 1]^d$ scale. All models are trained using Pytorch and our source code is publicly available[2].

We include the following algorithms for comparisons in attacking performance:

- Opt-based black-box attack (Opt-attack): our proposed algorithm.
- Boundary black-box attack (Brendel et al., 2017) (Boundary-attack): first work on attacking hard-label black box model in L2 distance metric. We use the authors' implementation and the parameters provided in Foolbox[3].

---

[1]https://github.com/pytorch/vision/tree/master/torchvision
[2]https://github.com/LeMinhThong/blackbox-attack
[3]https://github.com/bethgelab/foolbox

Table 1: Results for ($L_2$-norm based) untargeted attacks. ASR stands for Attack Success Rate.

| | MNIST | | CIFAR10 | | Imagenet (ResNet-50) | |
|---|---|---|---|---|---|---|
| | Avg $L_2$ | # queries | Avg $L_2$ | # queries | Avg $L_2$ | # queries |
| Boundary-attack | 1.1222 | 60,293 | 0.1575 | 123,879 | 5.9791 | 123,407 |
| | 1.1087 | 143,357 | 0.1501 | 220,144 | 3.7725 | 260,797 |
| Opt-attack | 1.188 | 22,940 | 0.2050 | 40,941 | 6.9796 | 71,100 |
| | 1.049 | 51,683 | 0.1625 | 77,327 | 4.7100 | 127,086 |
| | 1.011 | 126,486 | 0.1451 | 133,662 | 3.1120 | 237,342 |
| C&W (white-box) | 0.9921 | - | 0.1012 | - | 1.9365 | - |
| Random distortion | 1.0(2% ASR) | 300,000 | 0.15(10% ASR) | 300,000 | 3.11(0% ASR) | 480,000 |

- Limited black-box attack (Ilyas et al., 2018) (Limited-attack): previous work on targeted black-box attack in $L_\infty$ distance constraint in the hard-label setting. We use the authors' implementation and the parameters provided in Github[4].
- C&W white-box attack (Carlini & Wagner, 2017): one of the current state-of-the-art attacking algorithm in the white-box setting. We do binary search on parameter $c$ per image to achieve the best performance. Attacking in the white-box setting is a much easier problem, so we include C&W attack just for reference and indicate the best performance we can possibly achieve.
- Random distortion: we use 300,000 and 480,000 i.i.d random directions as the baseline. We note that while all other methods could achieve 100 % attack success rate (ASR), the ASR of random distortion appears to be quite low.

For all the cases except Limited-attack, we conduct adversarial attacks for randomly sampled $N = 100$ images from validation sets. Note that all three attacks (Opt-attack, Boundary-attack, C&W attack) have 100% successful rates, and we report the average $L_2$ distortion, defined by $\frac{1}{N} \sum_{i=1}^{N} \|x^{(i)} - x_0^{(i)}\|_2$, where $x^{(i)}$ is the adversarial example constructed by an attack algorithm and $x_0^{(i)}$ is the original $i$-th example. For black-box attack algorithms, we also report average and median number of queries for comparison. To compare with Limited-attack, we randomly sampled 50 images and report targeted ASR with average and median number of queries when limiting $L_\infty$ distortion to be 0.3 and 0.15 for ImageNet dataset. We also restrict the maximum number of queries to be 1,000,000 for all attacks.

### 4.1.1 UNTARGETED ATTACK

For untargeted attack, the goal is to turn a correctly classified image into any other label. The results are presented in Table 1. Note that for both Opt-attack and Boundary-attack, by changing the stopping conditions we can attain different distortions by varying the number of queries.

First, we compare Boundary-attack and the proposed Opt-attack in Table 1 and Table 5 in appendix. Our algorithm consistently achieves smaller distortion with less number of queries than Boundary-attack. For example, on MNIST data, we are able to reduce the number of queries by 3-4 folds, and Boundary-attack converges to worse solutions in all the 3 datasets. In addition, we include the average L2 norm distortion plot with different query budgets in Figure 5(a) in appendix. And our method consistently outperforms the Boundary-attack and achieves nearly 2x speedup on both datasets.

Compared with C&W attack, we found black-box attacks attain slightly worse distortion on MNIST and CIFAR. This is reasonable because white-box attack has much more information than black-box attack and is strictly easier. We note that the experiments in Brendel et al. (2017) conclude that C&W and Boundary-attack have similar performance because they only run C&W attack with a single regularization parameter $c$, without doing binary search to obtain the optimal parameter. For ImageNet, since we constraint the number of queries, the distortion of black-box attacks is much worse than C&W attack. The gap can be reduced by increasing the number of queries as showed in Figure 5(b) in appendix.

### 4.1.2 TARGETED ATTACK

The results for targeted attack is presented in Table 2 and Table 5 in appendix. Following the experiments in Brendel et al. (2017), for each randomly sampled image with label $i$ we set target label $t = (i + 1)$ module 10. On MNIST, we found our algorithm is more than 4 times faster (in terms of number of queries) than Boundary-attack and converges to a better solution. On CIFAR, our algorithm has similar efficiency with Boundary-attack at the first 60,000 queries, but converges to a slightly worse solution. Also, we show an example quality comparison from the same starting point to the original sample in Figure 4. And we also include some adversarial example in Figure 4.

---

[4]https://github.com/labsix/limited-blackbox-attacks

For attacks in $L_\infty$ norm constraint, we conduct experimental comparisons with Limited-attack[5]. the results are shown in Table 3. In the cases of an $L_\infty$ constraint $\epsilon \in \{0.15, 0.3\}$, our Opt-attack roughly halves the average number of queiries relative to Limited-attack. In addition, our ASR is 40% higher than Limited-attack when $\epsilon = 0.15$.

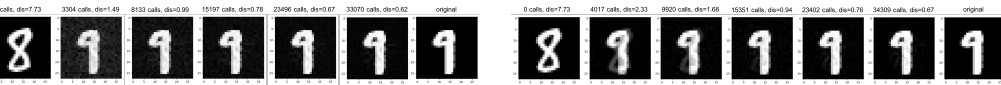

| (a) Examples of targeted Opt-attack | (b) Examples of targeted Boundary-attack |

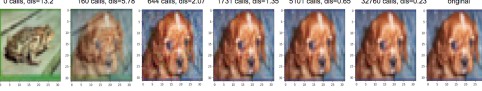
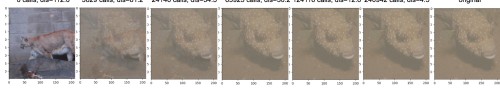

| (c) Examples of targeted Opt-attack on CIFAR-10 | (d) Examples of targeted Opt-attack on ImageNet |

Figure 4: (a)(b): Example quality comparison between targeted Opt-attack and Boundary-attack . Opt-attack can achieve a better result with less queries. (c)(d): Some adversarial examples generated by Opt-attack . From initialization image (left), through several number of model queries, we could generate a adversarial example very close to original image (right).

Table 2: Results of ($L_2$-norm based) targeted attack.

|  | MNIST | | CIFAR10 | |
|  | Avg $L_2$ | # queries | Avg $L_2$ | # queries |
|---|---|---|---|---|
| Boundary-attack (black-box) | 2.3158 | 30,103 | 0.2850 | 55,552 |
|  | 2.0052 | 58,508 | 0.2213 | 140,572 |
|  | 1.8668 | 192,018 | 0.2122 | 316,791 |
| Opt-attack (black-box) | 1.8522 | 46,248 | 0.2758 | 61,869 |
|  | 1.7744 | 57,741 | 0.2369 | 141,437 |
|  | 1.7114 | 73,293 | 0.2300 | 186,753 |
| C&W (white-box) | 1.4178 | - | 0.1901 | - |

Table 3: Results of targeted attacks in $L_\infty$ constraint. ASR stands for Attack Success Rate.

|  | $\epsilon$ | ASR | ImageNet (Inception V3) | |
|  |  |  | Avg queries | Median queries |
|---|---|---|---|---|
| Opt-attack | 0.30 | 100 % | 167,080 | 119,925 |
|  | 0.15 | 90% | 310,952 | 183,217 |
| Limited-attack | 0.30 | 100 % | 436,561 | 295,918 |
|  | 0.15 | 50% | 896,558 | 981,272 |

Table 4: Results of ($L_2$-norm based) untargeted attack on gradient boosting decision tree.

|  | HIGGS | | MNIST | |
|  | Avg $L_2$ | # queries | Avg $L_2$ | # queries |
|---|---|---|---|---|
| Opt-attack | 0.3458 | 4,229 | 0.6113 | 5,125 |
|  | 0.2179 | 11,139 | 0.5576 | 11,858 |
|  | 0.1704 | 29,598 | 0.5505 | 32,230 |

### 4.1.3 ATTACKING GRADIENT BOOSTING DECISION TREE (GBDT)

To evaluate our method's ability to attack models with discrete decision functions, we conduct our untargeted attack on gradient booting decision tree (GBDT). In this experiment, we use two standard datasets: HIGGS (Baldi et al., 2014) for binary classification and MNIST (LeCun et al., 1998) for multi-class classification. We use popular LightGBM framework to train the GBDT models and use suggested parameters in `https://github.com/Koziev/MNIST_Boosting`. To be more specific, for MNIST model, it has 100 trees and the max number of leaves in each tree is 100. For Higgs model, it has 255 trees and the max number of leaves in each tree is 500. And we don't limit the max depth on both models. We could achieve 0.8457 AUC for HIGGS and 98.09% accuracy for MNIST. The results of untargeted attack on GBDT are given in Table 4.

As shown in Table 4, by using around 30K queries, we could get a small distortion on both datasets, which firstly uncovers the vulnerability of GBDT models. Tree-based methods are well-known for its good interpretability. And because of that, they are widely used in the industry. However, we show that even with good interpretability and a similar prediction accuracy with convolution neural network, the GBDT models are vulnerable under our Opt-attack. This result raises a question about tree-based models' robustness, which will be an interesting direction in the future.

[5]Note that there's a bug in the original version of (Ilyas et al., 2018) for counting number of queries, which makes the reported number of queries less than they actually used. We use their modified version in our experiments, which has fixed this bug.

## 5 CONCLUSION

In this paper, we propose a generic and optimization-based hard-label black-box attack algorithm, which can be applied to discrete and non-continuous models other than neural networks, such as the gradient boosting decision tree. Our method enjoys query-efficiency and has a theoretical convergence guarantee on the attack performance under mild assumptions. Moreover, our attack achieves smaller or similar distortion using 3-4 times less queries compared with the state-of-the-art algorithms.

ACKNOWLEDGEMENT

We acknowledge the support by NSF IIS1719097, Intel faculty award, Google Cloud and AITRICS.

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

## 6 APPENDIX

### 6.1 MEDIAN RESULT

Table 5: Results of median L2 distortion on different attack methods.

| | Type | MNIST | | CIFAR10 | | Imagenet (ResNet-50) | |
| --- | --- | --- | --- | --- | --- | --- | --- |
| | | Median $L_2$ | # queries | Median $L_2$ | # queries | Median $L_2$ | # queries |
| Boundary-attack | Untargeted | 1.0832 | 142,686 | 0.1359 | 235,285 | 2.7879 | 261,258 |
| | Targeted | 1.8004 | 171,151 | 0.1962 | 314,839 | - | - |
| Opt-attack | Untargeted | 1.0206 | 127,077 | 0.1344 | 159,276 | 2.0687 | 246,755 |
| | Targeted | 1.7375 | 72,318 | 0.2020 | 158,438 | - | - |

### 6.2 RESULTS ON DIFFERENT NUMBER OF SAMPLE DIRECTIONS $u_t$

Table 6: Results of average L2 distortion on different number of sample directions $u_t$.

| | # of directions $u_t$ | MNIST | | | |
| --- | --- | --- | --- | --- | --- |
| | | Average $L_2$ | Average # queries | Median $L_2$ | Median # queries |
| Opt-attack | 1 | 2.0525 | 9,820 | 2.0188 | 8,093 |
| | 5 | 1.0550 | 59,901 | 1.0639 | 56,890 |
| | 10 | 1.0345 | 63,652 | 1.0420 | 62,209 |
| | 15 | 1.0257 | 71,045 | 1.0256 | 73,227 |
| | 20 | 1.0112 | 126,486 | 1.0206 | 127,077 |
| | 25 | 1.0098 | 146,516 | 1.0117 | 145,862 |

### 6.3 $L_2$ DISTORTION VERSUS MODEL QUERIES PLOT

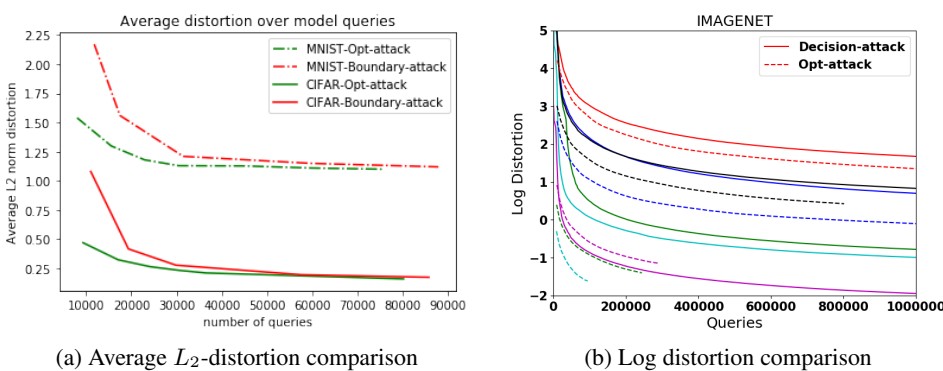

(a) Average $L_2$-distortion comparison    (b) Log distortion comparison

Figure 5: Left:Log distortion comparison of Boundary-attack (solid curves) vs Opt-attack (dotted curves) over number of queries for 6 different images. Right: Average L2-distortion versus number of queries plot.

### 6.4 THEORETICAL ANALYSIS

If $g(\boldsymbol{\theta})$ can be computed exactly, it has been proved in Nesterov & Spokoiny (2017) that RGF in Algorithm 2 requires at most $O(\frac{d}{\delta^2})$ iterations to converge to a point with $\|\nabla g(\boldsymbol{\theta})\|^2 \leq \delta^2$. However, in our algorithm the function value $g(\boldsymbol{\theta})$ cannot be computed exactly; instead, we compute it up to $\epsilon$-precision, and this precision can be controlled by binary threshold in Algorithm 1. We thus extend the proof in Nesterov & Spokoiny (2017) to include the case of approximate function value evaluation, as described in the following theorem.

**Theorem 1** *In Algorithm 2, suppose g has Lipschitz-continuous gradient with constant $L_1(g)$ and $g^*$ (optimal value) is finite. If the error of function value evaluation is controlled by $\epsilon = O(\beta\delta^2)$ and $\beta \leq \frac{\delta}{dL_1(g)}$, then in order to obtain $\frac{1}{N+1}\sum_{k=0}^{N} E_{\mathcal{U}_k}(||\nabla g(\boldsymbol{\theta}_k)||^2) \leq \delta^2$, the upper bound of total number of iterations is $O(\frac{d}{\delta^2})$.*

Note that the binary search procedure could obtain the desired function value precision in $O(\log \delta)$ steps. By using the same idea with Theorem 1 and following the proof in Nesterov & Spokoiny (2017), we could also achieve $O(\frac{d^2}{\delta^3})$ complexity when $g(\boldsymbol{\theta})$ is non-smooth but Lipschitz continuous.

Because there is a stopping criterion in Algorithm 1, we couldn't achieve the exact $g(\boldsymbol{\theta})$. Instead, we could get $\tilde{g}$ with $\epsilon$ error, i.e., $g(\boldsymbol{\theta}) - \epsilon \leq \tilde{g}(\boldsymbol{\theta}) \leq g(\boldsymbol{\theta}) + \epsilon$. Also, we define $\hat{\boldsymbol{g}}(\boldsymbol{\theta}) = \frac{\tilde{g}(\boldsymbol{\theta}+\beta\boldsymbol{u})-\tilde{g}(\boldsymbol{\theta})}{\beta} \cdot \boldsymbol{u}$ to be the noise gradient estimator.

Following Nesterov (2011), we define the Guassian smoothing approximation over $g(\theta)$, i.e,

$$g_\beta(\theta) = \frac{1}{\kappa} \int_E g(\theta + \beta u)e^{-\frac{1}{2}||u||^2}du. \tag{8}$$

Also, we have the upper bounds for the moments $M_p = \frac{1}{\kappa}\int_E ||u||^p e^{-\frac{1}{2}||u||^2}du$ from Nesterov (2011) Lemma 1.

For $p \in [0, 2]$, we have

$$M_p \leq d^{p/2}. \tag{9}$$

If $p \geq 2$, we have two-sided bounds

$$n^{p/2} \leq M_p \leq (p+n)^{p/2}. \tag{10}$$

## 6.5   PROOF OF THEOREM 1

Suppose f has a lipschitz-continuous gradient with constant $L_1(g)$, then

$$|g(y) - g(x) - \langle \nabla g(x), y - x\rangle| \leq \frac{1}{2}L_1(g)||x - y||^2 \tag{11}$$

We could bound $E_u(||\hat{g}(\boldsymbol{\theta})||^2)$ as follows,

Since
$$(\tilde{g}(\boldsymbol{\theta} + \beta u) - \tilde{g}(\boldsymbol{\theta}))^2 = [\tilde{g}(\boldsymbol{\theta} + \beta u) - \tilde{g}(\boldsymbol{\theta}) - \beta\langle\nabla g(\boldsymbol{\theta}), u\rangle + \beta\langle\nabla g(\boldsymbol{\theta}), u\rangle]^2$$
$$\leq 2(g(\boldsymbol{\theta} + \beta u) - g(\boldsymbol{\theta}) + \epsilon_{\boldsymbol{\theta}+\beta u} - \epsilon_{\boldsymbol{\theta}} - \beta\langle\nabla g(\boldsymbol{\theta}), u\rangle)^2 + 2\beta^2\langle\nabla g(\boldsymbol{\theta}), u\rangle^2 \tag{12}$$

Because $|\epsilon_{\boldsymbol{\theta}+\beta u} - \epsilon_{\boldsymbol{\theta}}| \leq 2\epsilon$,

$$[\tilde{g}(\boldsymbol{\theta} + \beta u) - \tilde{g}(\boldsymbol{\theta})]^2 \leq 2(\frac{\beta^2}{2}L_1(g)||u||^2)^2 + 4\beta^2 L_1(g)||u||^2\epsilon + 8\epsilon^2 + 2\beta^2\langle\nabla g(\boldsymbol{\theta}), u\rangle^2 \tag{13}$$

Take expectation over u, and with Theorem 3 in Nesterov (2011), which is $E_u(||g'(\boldsymbol{\theta}, u) \cdot u||^2) \leq (d+4)||\nabla g(\boldsymbol{\theta})||^2$

$$E_u(||\hat{g}(\boldsymbol{\theta})||^2) \leq \frac{\beta^2}{2}L_1^2(g)E_u(||u||^6) + 2E_u(||g'(\boldsymbol{\theta}, u) \cdot u||^2) + 4L_1(g)\epsilon E_u(||u||^4) + 8\frac{\epsilon^2}{\beta^2}E_u(||u||^2)$$

$$\leq \frac{\beta^2}{2}L_1^2(g)(d+6)^3 + 2(d+4)||\nabla g(\boldsymbol{\theta})||^2 + 4\epsilon L_1(g)(d+4)^2 + 8\frac{\epsilon^2}{\beta^2}d \tag{14}$$

With $\epsilon = O(\delta^2\beta)$, we could bound $E_u(||\tilde{g}(\boldsymbol{\theta})||^2)$

$$E_u(||\hat{g}(\boldsymbol{\theta})||^2) \leq \frac{\beta^2}{2}L_1^2(g)(d+6)^3 + 2(d+4)||\nabla g(\boldsymbol{\theta})||^2 + 4\beta L_1(g)(d+4)^2\delta^2 + 8d\delta^4 \tag{15}$$

And with

$$||\nabla g(\boldsymbol{\theta})||^2 \leq 2||\nabla g_\beta(\boldsymbol{\theta})||^2 + \frac{\beta^2}{2}L_1^2(g)(d+4)^2 \tag{16}$$

Which is proved in Nesterov (2011) Lemma 4.

Therefore, since $(n+6)^3 + 2(n+4)^3 \le 3(n+5)^3$, we could get

$$
\begin{aligned}
E_u(||\hat{g}(\boldsymbol{\theta})||^2) \le & \frac{\beta^2}{2} L_1^2(g)(d+6)^3 + 2(d+4)||\nabla g(\boldsymbol{\theta})||^2 + 2(d+4)||\nabla g(\boldsymbol{\theta})||^2 \\
& + 4\beta L_1(g)(d+4)^2\delta^2 + 8d\delta^4 \\
\le & \frac{\beta^2}{2} L_1^2(g)(d+6)^3 + 2(d+4)(2||\nabla g_\beta(\boldsymbol{\theta})||^2 + \frac{\beta^2}{2} L_1^2(g)(d+4)^2) \\
& + 4\beta L_1(g)(d+4)^2\delta^2 + 8d\delta^4 \\
\le & 4(d+4)||\nabla g_\beta(x)||^2 + \frac{3\beta^2}{2} L_1^2(g)(d+5)^3 + 4\beta L_1(g)(d+4)^2\delta^2 + 8d\delta^4
\end{aligned}
\tag{17}
$$

Therefore, since $g_\beta(\boldsymbol{\theta})$ has Lipshcitz-continuous gradient:

$$
|g_\beta(\boldsymbol{\theta}_{k+1}) - g_\beta(\boldsymbol{\theta}_k) + \alpha\langle\nabla g_\beta(\boldsymbol{\theta}_k), \hat{g}_\beta(\boldsymbol{\theta}_k)\rangle| \le \frac{1}{2}\alpha^2 L_1(g_\beta)||\hat{g}_\beta(\boldsymbol{\theta}_k)||^2
\tag{18}
$$

So that

$$
g_\beta(\boldsymbol{\theta}_{k+1}) \le g_\beta(\boldsymbol{\theta}_k) - \alpha\langle\nabla g_\beta(\boldsymbol{\theta}_k), \hat{g}_\beta(\boldsymbol{\theta}_k)\rangle + \frac{1}{2}\alpha^2 L_1(g_\beta)||\hat{g}_\beta(\boldsymbol{\theta}_k)||^2
\tag{19}
$$

Since

$$
\begin{aligned}
E_u(\hat{g}(\boldsymbol{\theta}_k)) &= \frac{1}{\kappa}\int_E \frac{g(\boldsymbol{\theta}+\beta u) - g(\boldsymbol{\theta}) + \epsilon_{\boldsymbol{\theta}+\beta u} - \epsilon_{\boldsymbol{\theta}}}{\beta} u e^{-\frac{1}{2}||u||^2} du \\
&= \nabla g_\beta(\boldsymbol{\theta}_k) + \frac{1}{\kappa}\int_E \frac{\epsilon_{\boldsymbol{\theta}+\beta u} - \epsilon_{\boldsymbol{\theta}}}{\beta} u e^{-\frac{1}{2}||u||^2} du \\
&\le \nabla g_\beta(\boldsymbol{\theta}_k) + \frac{2\epsilon}{\beta} n^{1/2} \cdot \mathbb{1}
\end{aligned}
\tag{20}
$$

where $\mathbb{1}$ is a all-one vector. Taking the expectation in $u_k$, we obtain

$$
E_{u_k}(g_\beta(\boldsymbol{\theta}_{k+1})) \le g_\beta(\boldsymbol{\theta}_k) - \alpha_k||\nabla g_\beta(\boldsymbol{\theta}_k)||^2 + \alpha_k\langle\nabla g_\beta(\boldsymbol{\theta}_k), \frac{2\epsilon}{\beta}d^{1/2}\cdot\mathbb{1}\rangle + \frac{1}{2}\alpha_k^2 L_1(g_\beta)E_{u_k}||\hat{g}_\beta(\boldsymbol{\theta}_k)||^2
$$

$$
\begin{aligned}
E_{u_k}(g_\beta(\boldsymbol{\theta}_{k+1})) \le & g_\beta(\boldsymbol{\theta}_k) - \alpha_k||\nabla g_\beta(\boldsymbol{\theta}_k)||^2 + \alpha_k\frac{2\epsilon}{\beta}n^{1/2}||\nabla g_\beta(\boldsymbol{\theta}_k)|| \\
& + \frac{1}{2}\alpha_k^2 L_1(g)(4(d+4)||\nabla g_\beta(\boldsymbol{\theta}_k)||^2 + \frac{3\beta^2}{2}L_1^2(g)(d+5)^3 + 4\beta L_1(g)(d+4)^2\delta^2 + 8d\delta^4)
\end{aligned}
\tag{21}
$$

Choosing $\alpha_k = \hat{\alpha} = \frac{1}{4(d+4)L_1(g)}$, we obtain

$$
\begin{aligned}
E_{u_k}(g_\beta(\boldsymbol{\theta}_k+1)) \le & g_\beta(\boldsymbol{\theta}_k) - \frac{1}{2}\hat{\alpha}||\nabla g_\beta(\boldsymbol{\theta}_k)||^2 + \hat{\alpha}\frac{2\epsilon}{\beta}d^{1/2}||\nabla g_\beta(\boldsymbol{\theta}_k)|| + \frac{3\beta^2}{64}L_1(g)\frac{(d+5)^3}{(d+4)^2} \\
& + \frac{\beta}{8}\delta^2 + \frac{d}{4(d+4)^2 L_1(g)}\delta^4
\end{aligned}
\tag{22}
$$

Since $(d+5)^3 \le (d+8)(d+4)^2$, taking expectation over $\mathcal{U}_k$, where $\mathcal{U}_k = \{u_1, u_2, \ldots, u_k\}$, we get

$$
\phi_{k+1} \le \phi_k - \frac{1}{2}\hat{\alpha}E_{\mathcal{U}_k}(||\nabla g_\beta(\boldsymbol{\theta}_k)||^2) + \frac{3\beta^2(d+8)}{64}L_1(g) + \frac{\beta}{8}\delta^2 + \frac{d}{4(d+4)^2 L_1(g)}\delta^4 + \hat{\alpha}d^{1/2}E_{\mathcal{U}_k}(||\nabla g_\beta(\boldsymbol{\theta}_k)||)\delta^2
\tag{23}
$$

Where $\phi_k = E_{\mathcal{U}_{k-1}(g(\boldsymbol{\theta}_k))}, k \ge 1$ and $\phi_0 = g(\boldsymbol{\theta}_0)$.

Assuming $g(x) \ge g^*$, summing over k and divided by N+1, we get

$$
\begin{aligned}
\frac{1}{N+1}\sum_{k=0}^N E_{\mathcal{U}_k}(||\nabla g_\beta(\boldsymbol{\theta}_k)||^2) \le & 8(d+4)L_1(g)[\frac{g(x_0) - g^*}{N+1} + \frac{3\beta^2(d+8)}{16}L_1(g) + \frac{\beta}{8}\delta^2 \\
& + \frac{d}{4(d+4)^2 L_1(g)}\delta^4 + \frac{1}{N+1}\sum_{k=0}^N E_{\mathcal{U}_k}(||\nabla g_\beta(\boldsymbol{\theta}_k)||)\delta^2]
\end{aligned}
\tag{24}
$$

Clearly, $\frac{1}{N+1}\sum_{k=0}^N E_{\mathcal{U}_k}(||\nabla g_\beta(\boldsymbol{\theta}_k)||) \le \delta^2$.

Since $\vartheta_k^2 = E_{\mathcal{U}_k}(||\nabla g(\boldsymbol{\theta}_k)||^2) \leq 2E_{\mathcal{U}_k}(||\nabla g_\beta(\boldsymbol{\theta}_k)||^2) + \frac{\beta^2(d+4)^2}{2}L_1^2(g)$, $\vartheta_k^2$ is in the same order of $E_{\mathcal{U}_k}(||\nabla g_\beta(\boldsymbol{\theta}_k)||^2)$. In order to get $\frac{1}{N+1}\sum_{k=0}^{N}\vartheta_k^2 \leq \delta^2$, we need to choose $\beta \leq \frac{\delta}{dL_1(g)}$, then N is bounded by $O(\frac{d}{\delta^2})$

