# OpenReview forum: "Query-Efficient Hard-label Black-box Attack: An Optimization-based Approach"
_ICLR.cc/2019/Conference_

### Official Review · AnonReviewer2 · 2018-11-04
**well-written paper with good empirical results**

**Rating:** 7
**Confidence:** 4

**Review:**

This paper addresses black-box classifier attacks in the “hard-label” setting, meaning that the only information the attacker has access to is single top-1 label predictions. Relative to even the standard black-box setting where the attacker has access to the per-class logits or probabilities, this setting is difficult as it makes the optimization landscape non-smooth. The proposed approach reformulates the optimization problem such that the outer-loop optimizes the direction using approximate gradients, and the inner-loop estimates the distance to the nearest attack in a given direction. The results show that the proposed approach successfully finds both untargeted and targeted adversarial examples for classifiers of various image datasets (including ImageNet), usually with substantially better query-efficiency and better final results (lower distance and/or higher success rate) than competing methods.

=====================================

Pros:

Very well-written and readable paper with good background and context for those (like me) who don’t closely follow the literature on adversarial attacks. Figs. 1-3 are nice visual aids for understanding the problem and optimization landscape.

Novel formulation and approach that appears to be well-motivated from the literature on randomized gradient-free search methods. Novel theoretical analysis in Appendix that generalizes prior work to approximations (although, see notes below).

Good empirical results showing that the method is capable of query-efficiently finding attacks of classifiers on real-world datasets including ImageNet. Also shows that the model needn’t be differentiable to be subject to such attacks by demonstrating the approach on a decision-tree based classifier. Appears to compare to and usually outperform appropriate baselines from prior work (though I’m not very familiar with the literature here).

=====================================

Cons/questions/suggestions/nitpicks:

Eq 4-5: why \texttt argmin? Inconsistent with other min/maxes.

Eq 4-5: Though I understood the intention, I think the equations are incorrect as written: argmin_{\lambda} { F(\lambda) } of a binary-valued function F would produce the set of all \lambdas that make F=0, rather than the smallest \lambda that makes F=1. I think it should be something like:

min_{\lambda>0} {\lambda}
s.t. f(x_0+\lambda \theta/||\theta||) != y_0

Sec 3.1: why call the first search “fine-grained”? Isn’t the binary search more fine-grained? I’d suggest changing it to “coarse-grained” unless I’m misunderstanding something.

Algorithm 2: it would be nice if this included all the tricks described as “implementation details” in the paragraph right before Sec. 4 -- e.g. averaging over multiple sampled directions u_t and line search to choose the step size \eta. These seem important and not necessarily obvious to me.

Algorithm 2: it could be interesting to show how performance varies with number of sampled directions per step u_t.

Sec: 4.1.2: why might your algorithm perform worse than boundary-attack on targeted attacks for CIFAR classifiers? Would like to have seen at least a hypothesis on this.

Sec 6.3 Theorem 1: I think the theorem statement is a bit imprecise. There is an abuse of big-O notation here -- O(f(n)) is a set, not a quantity, so statements such as \epsilon ~ O(...) and \beta <= O(...) and “at most O(...)” are not well-defined (though common in informal settings) and the latter two are redundant given the meaning of O as an upper bound. The original theorem from [Nesterov & Spokoiny 2017] that this Theorem 1 would generalize doesn’t rely on big-O notation -- I think following the same conventions here might improve the theorem and proof.

=====================================

Overall, this is a good paper with nice exposition, addressing a challenging but practically useful problem setting and proposing a novel and well-motivated approach with strong empirical results.

---

> ### Author Response · Authors · 2018-11-14
> **Thanks for the great suggestions and we have revised our paper as follows:**
>
> Thanks for the positive reviews and valuable suggestions.
>
> 1. We have modified equation (4-5) to min_{\lambda>0} {\lambda} s.t. f(x_0+\lambda \theta/||\theta||) != y_0  as you suggested.
>
> 2. Section 3.1: Yes, you are right. We have followed your suggestion by changing “fine-grained” to “coarse-grained” in the revision.
>
> 3. Algorithm 2:
> A. We have included all implementation details above section 4 following your suggestion in revision.
> B. We have added a new table to show how the performance varies with the number of sampled directions per step u_t in Appendix 6.2.
>
> 4. The performance in CIFAR10: During the experiment, we found that CIFAR is more sensitive to the initial direction in our method. Although we find a relatively small distortion direction at first, the method sometimes converges to a worse point than Boundary-attack. It could be solved by selecting several directions as initialization and do Algorithm 2 several times.
>
> 5. The big O notation in proof: We have followed your suggestion to replace $\epsilon\simO()$ to $\epsilon=O()$ and delete the big O notation with \beta.

---

### Official Review · AnonReviewer3 · 2018-11-05
**Interesting paper**

**Rating:** 6
**Confidence:** 5

**Review:**

This paper proposed a reformulation of objective function to solve the hard-label black-box attack problem. The idea is interesting and the performance of the proposed method seem to be capable of finding adversarial examples with smaller distortions and less queries compared with other hard-label attack algorithms.

This paper is well-written and clear.

==============================================================================================
Questions

A. Can it be proved the g(theta) is continuous? Also, the theoretical analysis assume the property of Lipschitz-smooth and thus obtain the complexity of number of queries. Does this assumption truly hold for g(theta), when f is a neural network classifiers? If so, how to obtain the Lipschitz constant of g that is used in the analysis sec 6.3?

B. What is the random distortion in Table 1? What initialization technique is used for the query direction in the experiments?

C. The GBDT result on MNIST dataset is interesting. The authors should provide tree models description in 4.1.3. However, on larger dataset, say imagenet, are the tree models performance truly comparable to ImageNet? If the test accuracy is low, then it seems less meaningful to compare the adversarial distortion with that of imagenet neural network classifiers. Please explain.

D. For sec 3.2, it is not clear why the approximation is needed. Because the gradient of g with respect to theta is using equation (7) and theta is already given (from sampling); thus the Linf norm of theta is a constant. Why do we need the approximation? Given that, will there be any problems on the L2 norm case?

---

> ### Author Response · Authors · 2018-11-14
> **Thanks for the review and we have answered your questions as follows**
>
> 1. Without additional assumptions, we couldn’t prove g(theta) is continuous for general deep neural networks. It’s true that the g(theta) may not be continuous; for example, we think it might be possible to construct some counter-examples using ReLU activation. However, although the assumption may not hold for DNN or GBDT globally, our algorithm still performs well in practice. Moreover, we have illustrated that the decision boundaries of DNN and GBDT are mostly smooth (Fig2,3) in some examples. What we can assure is that if $g(\cdot)$ has Lipschitz continuous gradient, our algorithm has such a theoretical guarantee. Moreover, based on the same analysis in section 7 of [21], the condition can be relaxed to Lipschitz continuous. This is indeed a sufficient but not necessary condition.
>
> 2. We use random directions instead of any format of attack to generate adversarial examples. More specifically, we generate i.i.d. random directions \theta_1, … \theta_n from Gaussian, and  for each of them check whether it’s successful or not (successful if $f(x_i+\theta)\neq y_i$). We have added more details in the revised paper.
>
> 3.  We have provided the tree models description in 4.1.3.
>
> We don’t really know any tree-based model that can achieve similar performance with CNN on ImageNet, but GBDT is still useful for many real-world data science applications (e.g., it’s the most common model for click-through rate predictions). We would like to stress that it’s not our focus to discuss whether GBDT is useful or not. The aim of attacking GBDT is to prove that our algorithm can also be used to attack other discrete and non-continuous machine learning models, which couldn’t be done by current gradient-based attack methods.
>
> 4. About the approximation of L-inf norm: yes, we could directly apply opt-attack on L-inf norm without any modification. However, we find it harder to optimize in practice because of the additional max term. With the approximation, the function of g is more smooth than previous, which leads to faster convergence.

---

### Official Review · AnonReviewer1 · 2018-11-09
**Nice idea with solid experiments**

**Rating:** 7
**Confidence:** 3

**Review:**

In this paper the authors propose optimizing for adversarial examples against black box models by considering minimizing the distance to the decision boundary.  They show that because this gives real valued feedback, the optimizer is able to find closer adversarial examples with fewer queries.  This would be heavily dependent on the model structure (with more complex decision boundaries likely being harder to optimize) but they show empirically in 4 models that this method works well.

I am not convinced that the black box model setting is the most relevant (and 20k queries is still a fair bit), but this is important research nonetheless.  I generally found the writing to be clear and the idea to be elegant; I think readers will value this paper.

---

> ### Author Response · Authors · 2018-11-14
> **Thanks for the positive review and we have some clarifications**
>
> Thanks for the positive reviews. We agree that white-box setting could be a better way to evaluate the model’s robustness. However, if an attacker wants to attack a system in the real world, it’s usually in the black-box setting. In practice, commercial systems like Google Cloud vision only output the top-1 or top-k predictions to users, which is the same with our hard-label black-box setting, where no model information is revealed except that the attacker can make queries to probe the corresponding hard-label decisions. We agree that 20k queries are still too much and how to reduce the number of queries is still an open and challenging problem.

---

### Meta-Review · Area_Chair1 · 2018-12-17
**Good paper, accept**

**Confidence:** 5
**Recommendation:** Accept (Poster)

**Metareview:**

The reviewers liked the clarity of the material and agreed the experimental study is convincing. Accept.